# Dynamics of Micronutrient Uptake and Removal by Three Modern Runner Peanut Cultivars

Carlos Alexandre Costa Crusciol [1,*], José Roberto Portugal [1], João William Bossolani [1],
Luiz Gustavo Moretti [1], Adalton Mazetti Fernandes [2], Adônis Moreira [3], Jader Luis Nantes Garcia [4],
Gleize Leviski de Brito Garcia [4], Cristiane Pilon [5] and Heitor Cantarella [6,*]

[1] Department of Crop Science, College of Agricultural Sciences, São Paulo State University (UNESP),
Botucatu 18610-034, SP, Brazil
[2] Center of Research for Tropical Tubers and Starches (CERAT), São Paulo State University (UNESP),
Botucatu 18610-034, SP, Brazil
[3] Embrapa Soybean, Carlos João Strass Highway, Londrina 86085-981, PR, Brazil
[4] Department of Forest, Soil and Environmental Sciences, College of Agricultural Sciences,
São Paulo State University (UNESP), Botucatu 18610-034, SP, Brazil
[5] College of Agricultural and Environmental Science, University of Georgia, 2356 Rainwater Road,
Tifton, GA 31793, USA
[6] Agronomic Institute of Campinas (IAC), Soils and Environmental Resources Center,
Av. Barão de Itapura 1481, Campinas 13020-902, SP, Brazil
* Correspondence: carlos.crusciol@unesp.br (C.A.C.C.); cantarella@iac.sp.gov.br (H.C.)

**Abstract:** Micronutrient fertilization is usually neglected by producers, especially for peanut, a crop that is frequently grown in crop rotation systems due to its low perceived nutrient requirements. New peanut cultivars are able to achieve high yields when grown under suitable conditions. However, fertilization recommendation tables are dated and do not consider the need for micronutrients. To support improvements in these recommendations, this study quantified the micronutrient demand (B, Cu, Fe, Mn, and Zn) of three runner peanut cultivars (IAC Runner 886, IAC 505, and IAC OL3) during the biological cycle and the transport of these micronutrients to pods and kernels. The experiment was carried out in a randomized complete block with a split-plot design and nine replications. The whole plots consisted of the three peanut cultivars, and the subplots comprised nine plant samplings (at 14, 28, 42, 56, 70, 84, 105, 126, and 147 days after emergence (DAE)). These modern peanut cultivars exhibited high uptake and accumulation of Fe, but the proportion of Fe removed by pods and kernels was lowest among all analyzed micronutrients. The second-most-accumulated micronutrient was Mn. The maximum requirement for micronutrients of peanut occurred around 84 DAE, and IAC 505 had the highest micronutrient uptake and accumulation among the cultivars (especially at later stages), followed by IAC OL3 and IAC Runner 886. Our results provide new insights into micronutrient requirements for peanut and demonstrate the need for new fertilizer recommendation programs for peanut cultivation.

**Keywords:** agricultural systems; nutrient management; peanut; plant nutrition; soil fertility and productivity

## 1. Introduction

Plants are an important source of micronutrients for human health [1,2]. Among cultivated plant species, peanut (*Arachis hypogaea* L.) is an excellent source of proteins and lipids and is therefore one of the main oilseed crops cultivated worldwide [3]. Since 2016, China has been the largest producer of peanut, with approximately 17 million tons per year, corresponding to ~36% of the total volume produced worldwide [4]. Another major peanut producer is Brazil, which has increased peanut production to approximately 570 thousand tons per year [5]. Peanut cultivars belonging to the Runner group are widely cultivated in the United States and Brazil [6,7]. Other commonly cultivated varieties belong to the

Spanish and Valencia groups [6]. In recent years, the production of modern cultivars, also known as high oleic cultivars, has expanded considerably [8].

The major driver of increased peanut production in Brazil is the use of this crop in rotation, especially after sugarcane (*Saccharum* sp.) or pasture [9], which deplete nutrients from the soil [10]. In these cropping systems, peanut serves as a plant conditioner to increase soil fertility before cane field renovation and reduce sugarcane renovation costs, which are quite high for this energy crop [11,12]. Due to their high biological nitrogen fixation (BNF) efficiency, plants in the legume family such as peanut have low demand for pesticides and fertilizers, especially nitrogen (N) [13,14]. Consequently, the real fertilizer requirements for peanut to reach its maximum productive potential have received little attention compared with other cultivated species [15,16], and most producers neglect fertilizer requirements for this crop [11]. In addition, peanut is usually cultivated in sandy soils with low organic matter content [17], which favors micronutrient deficiency [18]. Neglecting the demand of peanut for micronutrients during the crop's biological cycle may have even graver consequences than overlooking primary macronutrients, reinforcing the need for micronutrient-fertilizer programs based on periods of greater demand for peanut.

Among micronutrients, much is known about the requirements for molybdenum (Mo) and cobalt (Co) in BNF in legumes [19], but other micronutrients that participate in this process have rarely been considered. Iron (Fe) is essential for several key enzymes of the nitrogenase complex in BNF, such as iron-sulfur (FeS) clusters, which contain a FeMoCo cofactor at the active site for $N_2$ reduction [20]. Fe is also required for the electron carrier ferredoxin and as a cofactor in the formation of leghemoglobin, an oxygen-binding protein, in vegetables [21]. Another important micronutrient in BNF is boron (B). Leguminous plants with a low B supply exhibit dramatic changes in $N_2$ fixation [22–24]. B deficiency reduces the level of infection of host cells by *Rhizobia* [25], and causes morphological aberrations [26] that decrease the efficiency of the barriers responsible for preventing $O_2$ diffusion in nodules, leading to nitrogenase inactivation and reduced N fixation [27]. Increasing the BNF efficiency of peanut is very important because all of the plant's demand for N is met by this process. Therefore, it is essential to establish the demand for micronutrients throughout the peanut biological cycle.

In addition to BNF, numerous other metabolic processes in plants are mediated by micronutrients [28]. The synthesis of hormones and other substances related to plant growth and development is highly dependent on manganese (Mn), Zn and B [29]. Antioxidant enzymes, which reduce oxidative damage, are activated by Fe, Zn, Mn and copper (Cu) [29]. Zn is also required for the synthesis of amino acids and vitamins, while Cu and Fe are of paramount importance in respiration [29].

Micronutrient fertilization recommendations for peanut are scarce. Studies have shown responses of peanut to, for example, leaf application of boron (B) at doses between 1.0 and 1.5 kg ha$^{-1}$ split into three applications [30], leaf application of manganese (Mn) at 0.3 to 0.6 kg ha$^{-1}$ for each of three to four applications [31], and the application of zinc (Zn) in soil at 8.0 kg ha$^{-1}$ [32] or on leaves at 0.5 kg ha$^{-1}$ [33]. Moreover, the literature is silent on recommendations for supplying peanut with Cu and Fe. These gaps reinforce the need for research to support the development of micronutrient-fertilizer programs based on periods of greater demand for peanut that will allow this leguminous plant to reach high productivity levels [29]. To contribute to addressing this gap, the present study quantified the demand for micronutrients (B, Cu, Fe, Mn, and Zn) during the biological cycle of three runner peanut cultivars (IAC Runner 886, IAC 505, and IAC OL3) widely grown in Brazil.

## 2. Materials and Methods

### 2.1. Site, Climate, and Soil

The site description, soil properties, experimental design, peanut cultivar characteristics, and crop management are briefly presented here. The full details can be found in [34]. The field experiment was conducted in Botucatu, São Paulo, in southeastern Brazil (48°23′ W, 22°51′ S; WGS84; 765 m a.s.l.) during the 2014/2015 growing season in a Red Latosol [35]

or thermic Dystroferric Red Oxisol [36]. The soil in the experimental area had the following properties at a depth of 0.00–0.20 m before peanut sowing: pH(CaCl$_2$) = 5.9; soil organic matter = 30 g dm$^{-3}$; P (resin) = 60 mg dm$^{-3}$; K$^+$ = 7.7 mmol$_c$ dm$^{-3}$; Ca$^{2+}$ = 59 mmol$_c$ dm$^{-3}$; Mg$^{2+}$ = 38 mmol$_c$ dm$^{-3}$; H + Al = 26 mmol$_c$ dm$^{-3}$; CEC = 131 mmol$_c$ dm$^{-3}$; base saturation = 80%; SO$_4$-S= 24 mg dm$^{-3}$; B = 0.36 mg dm$^{-3}$; Cu = 12.6 mg dm$^{-3}$; Fe = 16.0 mg dm$^{-3}$; Mn = 34.4 mg dm$^{-3}$; and Zn = 2.6 mg dm$^{-3}$. The 50-year rainfall average is 1360 mm year$^{-1}$, and the mean annual air temperature is 20.7 °C [37]. Meteorological data during the growing season are shown in Table S1.

The experiment was arranged In a randomized complete block design with split-plots and nine replications. The plots consisted of three peanut cultivars, IAC Runner 886, IAC 505, and IAC OL3, and the split-plots consisted of nine plant samplings (assessments), which occurred at 14, 28, 42, 56, 70, 84, 105, 126, and 147 days after emergence (DAE). Information about the size, shape, and useful area of the plots and split-plots and the peanut cultivars is reported in [34].

Peanut sowing was performed mechanically on 8 December 2014, at a row spacing of 0.90 m and seeding rate of 14 seeds m$^{-1}$. Mineral fertilization was applied in the furrow as 20 kg ha$^{-1}$ N, 70 kg ha$^{-1}$ P$_2$O$_5$, and 40 kg ha$^{-1}$ K$_2$O using NPK fertilizer 08-28-16. No S or micronutrients were applied. Seedling emergence occurred 10 days after sowing (DAS), when more than 50% of the seedlings within each plot emerged from the soil. Information about weed management and recommended agricultural practices performed in the field is provided in detail in [34].

### 2.2. Plant Measurements and Analysis

A complete description of plant sampling to evaluate plant nutritional status and obtain dry matter (DM) data is provided in [34]. The contents of B, Cu, Fe, Mn, and Zn were determined in dry samples of the apical cluster of the main branch collected at the full-bloom stage and in dry samples of each plant tissue collected at each plant sampling (assessment) according to the methodology proposed in [38]. Micronutrients were extracted by nitric acid–perchloric acid wet digestion, and determined by atomic absorption spectrophotometry. Based on the micronutrient content in the plant tissues and the dry matter (DM) accumulation reported in [34], the accumulation of micronutrients in each plant tissue was calculated. The accumulation rates of micronutrients in the whole plant and the amounts of these nutrients taken up per ton of pods or kernels produced were calculated according to the protocol documented for macronutrients in [34].

### 2.3. Pod and Kernel Yields and Nutrient Removal

Details on the harvest and calculation of the yields of pods and kernels can be found in [34]. Samples of pods and kernels from each subplot were dried in a forced-air oven at 65 °C for 72 h and ground to determine the micronutrient content (B, Cu, Fe, Mn, and Zn) [38]. The micronutrient removal per area or per ton of pods/kernels and the relative removal of these nutrients were calculated as documented for macronutrients in [34].

### 2.4. Statistical Analysis

The data were subjected to (analysis of variance) ANOVA. The means for cultivars at each sampling time were separated by Fisher's protected least significant difference (LSD) test at 0.05 probability. The effects of plant sampling on nutrient accumulation variables were assessed by regression analysis using SigmaPlot 10.0 software.

## 3. Results

### Leaf Content and Micronutrient Uptake

The contents of B and Zn in leaves did not differ among the three cultivars (Table 1). By contrast, the Cu content was higher in IAC Runner than in IAC OL3, and Fe and Mn content were highest in IAC 505.

**Table 1.** Contents of B, Cu, Fe, Mn, and Zn in leaves of three peanut cultivars at the full-bloom stage.

| Cultivars | B | Cu | Fe | Mn | Zn |
|---|---|---|---|---|---|
| | | | mg kg$^{-1}$ | | |
| IAC Runner | 52 ± 4.3 a | 20 ± 1.3 a | 193 ± 8.1 b | 95 ± 5.9 b | 45 ± 1.9 a |
| IAC 505 | 60 ± 4.2 a | 19 ± 1.1 ab | 270 ± 7.2 a | 109 ± 7.1 a | 47 ± 1.6 a |
| IAC OL3 | 54 ± 3.8 a | 17 ± 0.9 b | 178 ± 6.9 b | 100 ± 5.4 ab | 47 ± 1.5 a |
| CV% | 12.3 | 6.5 | 7.9 | 5.9 | 13.4 |

Values ± SE (standard error) followed by the same letter in the same column are not significantly different at $p \leq 0.05$, according to the LSD test.

In all peanut cultivars, B accumulation in leaves was highest around 90 DAE (Figure 1a). Between 70 and 126 DAE, IAC OL3 exhibited the highest B accumulation in leaves. B accumulation in stems peaked first in IAC Runner (99 DAE) and then in IAC OL3 (109 DAE) and IAC 505 (114 DAE) (Figure 1b). At 147 DAE, IAC Runner exhibited lower B accumulation in stems than the other two cultivars. In reproductive structures, the peak of B accumulation occurred around 104 DAE in IAC Runner and IAC OL3, but at around 128 DAE in IAC 505 (Figure 1c). IAC 505 had a greater accumulation of B in reproductive structures than the other cultivars at 70, 126, and 147 DAE. IAC 505 and IAC Runner showed a rapid accumulation of B in pods between 84 and 105 DAE (Figure 1d). After 105 DAE, IAC 505 accumulated the most B in pods among the cultivars, and at the end of the cycle, this accumulation was approximately 40% greater in IAC 505 than in the other cultivars. The peak of B accumulation in the whole plant occurred around 106 DAE in IAC Runner and IAC OL3, both of which had B accumulation of approximately 306 g ha$^{-1}$, but it was at around 118 DAE in IAC 505 (349 g ha$^{-1}$) (Figure 1e). Between 105 and 147 DAE, IAC 505 outperformed the other cultivars in B accumulation in the whole plant. For all cultivars, the maximum rate of B accumulation in the whole plant was 4.5 g ha$^{-1}$ d$^{-1}$ around 70 DAE (Figure 1f).

Regardless of the peanut cultivar, the maximum accumulation of Cu in leaves (53 g ha$^{-1}$) occurred around 86 DAE (Figure 2a). In plant stems, peak Cu accumulation occurred around 94 DAE in all cultivars (Figure 2b). Between 70 and 147 DAE, the Cu accumulation in stems was greatest in IAC 505. The maximum Cu accumulation in reproductive structures occurred later in IAC 505 (120 DAE; 12 g ha$^{-1}$) than in IAC Runner (105 DAE; 10 g ha$^{-1}$) and IAC OL3 (105 DAE; 14 g ha$^{-1}$) (Figure 2c). At 147 DAE, IAC 505 had accumulated two-fold more Cu in reproductive structures than the other cultivars. In the whole plant, the maximum accumulation of Cu occurred around 116 DAE in IAC Runner and IAC OL3 and 130 DAE in IAC 505 (Figure 2e). Between 105 and 147 DAE, Cu accumulation in the whole plant was greatest in IAC 505. The maximum rate of Cu accumulation in the whole plant occurred at 70 DAE (1.9 g ha$^{-1}$ d$^{-1}$) in IAC Runner and IAC OL3, but at around 80 DAE in IAC 505 (2.2 g ha$^{-1}$ d$^{-1}$) (Figure 2f).

For Fe, the accumulation in leaves peaked at approximately 110 DAE for the three peanut cultivars and ranged from 418 g ha$^{-1}$ in IAC Runner and IAC 505 to 549 g ha$^{-1}$ in IAC OL3 (Figure 3a). Fe accumulation in stems was greatest around 119 DAE in IAC Runner and IAC OL3 and 136 DAE in IAC 505 (Figure 3b). In all cultivars, Fe accumulation in stems increased throughout the season; however, the differences in Fe accumulation among the cultivars was most pronounced at 147 DAE, with the greatest accumulation in IAC 505. In reproductive structures, the greatest Fe accumulation occurred at approximately 110 DAE in IAC Runner and IAC OL3, nine days earlier than in IAC 505 (Figure 3c). The greatest Fe accumulation in pods occurred at 147 DAE in all cultivars, with IAC 505 accumulating more Fe than the other two cultivars (Figure 3d). Peak Fe accumulation in the whole plant was observed at approximately 115 DAE in IAC Runner and IAC OL3, but at approximately 124 DAE in IAC 505 (Figure 3e). At the end of the crop season (147 DAE), Fe accumulation in the whole plant was highest in IAC 505. The peak rate of Fe accumulation in the whole plant occurred at approximately 84 DAE in all cultivars, but was highest in IAC OL3 (Figure 3f).

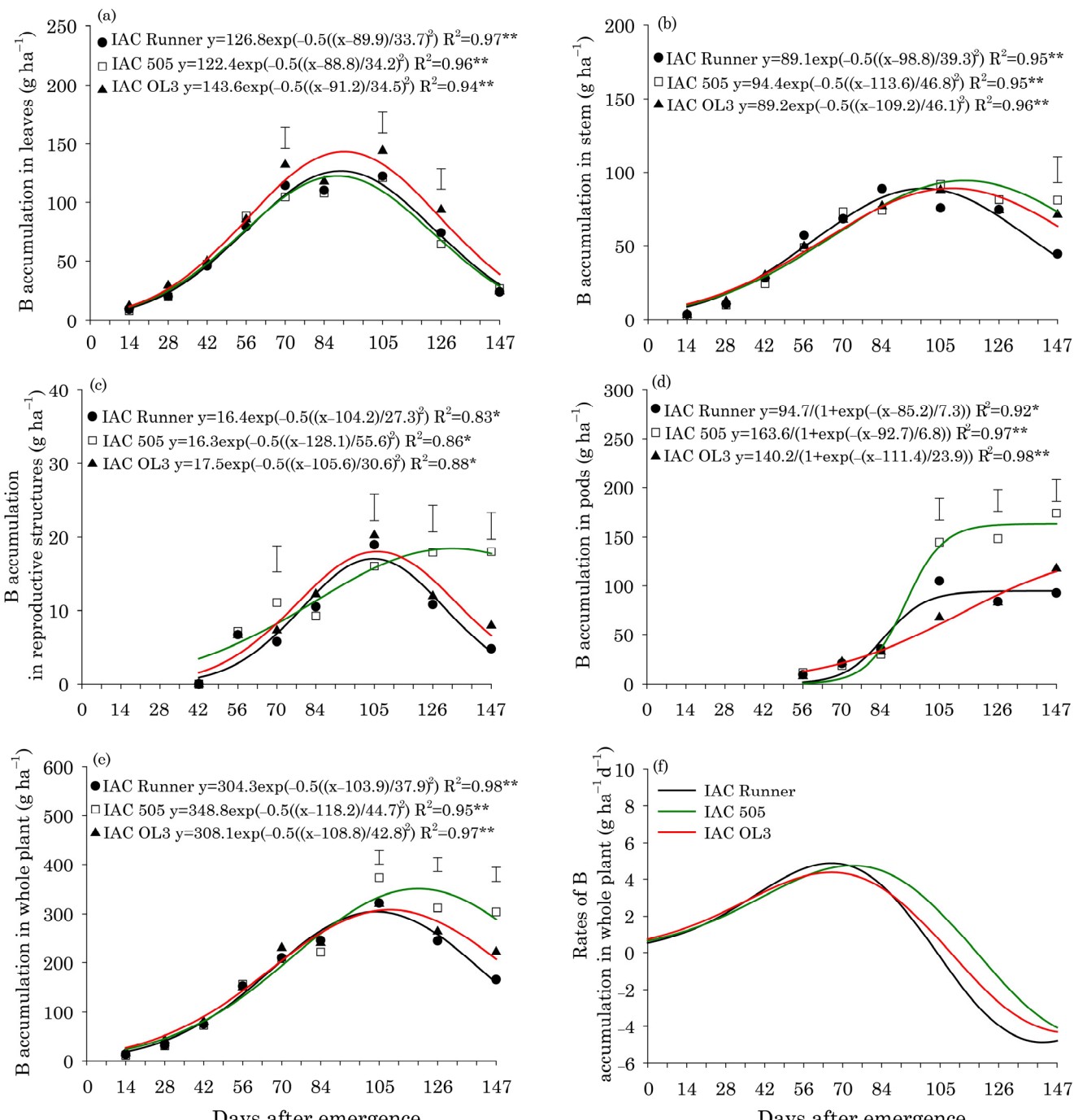

**Figure 1.** Boron (B) accumulation in leaves (**a**), stems (**b**), reproductive structures (**c**), pods (**d**), and the whole plant (**e**), and B accumulation rates in the whole plant (**f**) of three peanut cultivars throughout the season. ** and * indicate significance at $p \leq 0.01$ and $p \leq 0.05$, respectively, by the F test. The vertical bars indicate the least significant difference values by the LSD test at $p \leq 0.05$.

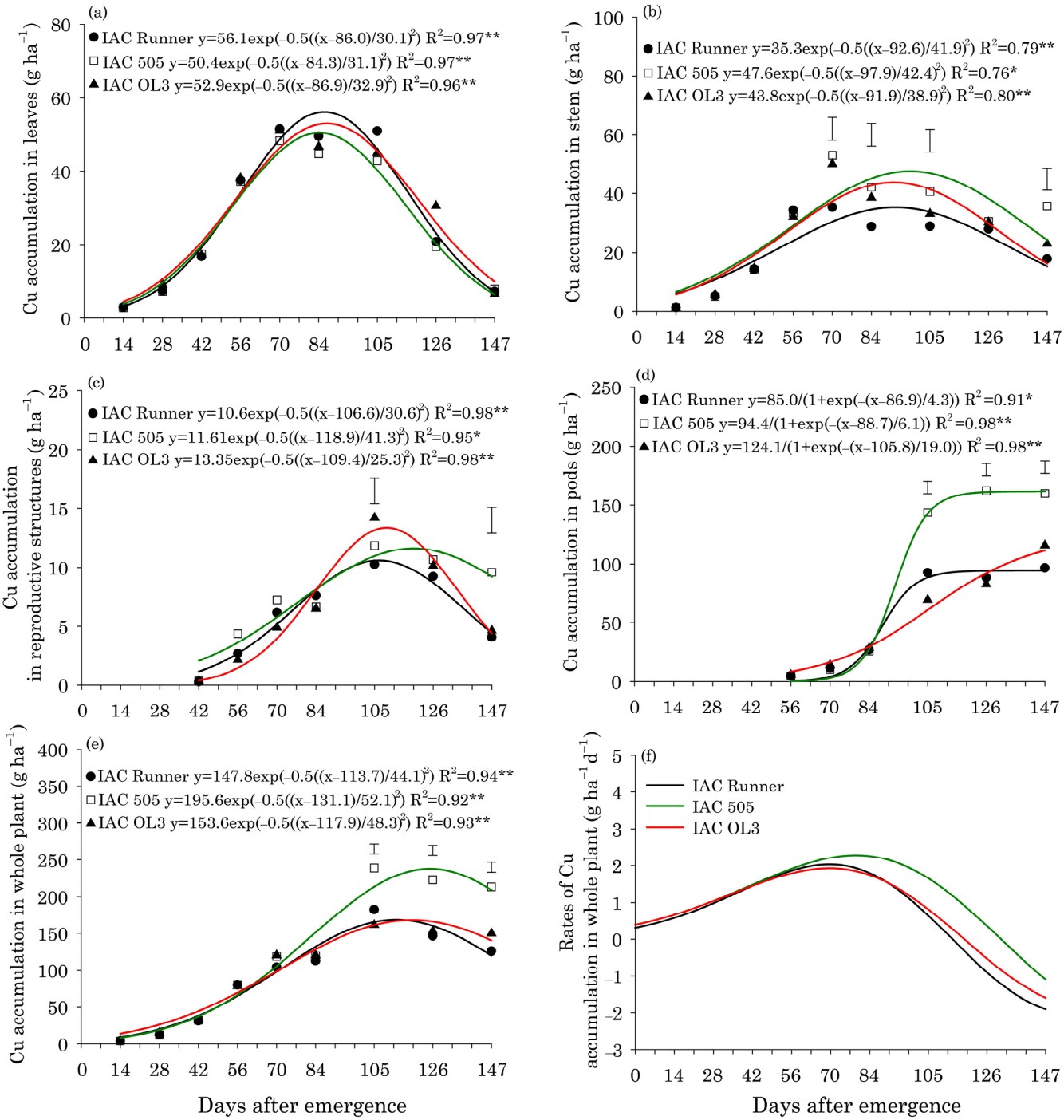

**Figure 2.** Copper (Cu) accumulation in leaves (**a**), stems (**b**), reproductive structures (**c**), pods (**d**), and the whole plant (**e**), and Cu accumulation rates in the whole plant (**f**) of three peanut cultivars throughout the season. ** and * indicate significance at $p \leq 0.01$ and $p \leq 0.05$ by the F test. The vertical bars indicate the least significant difference values by the LSD test at $p \leq 0.05$.

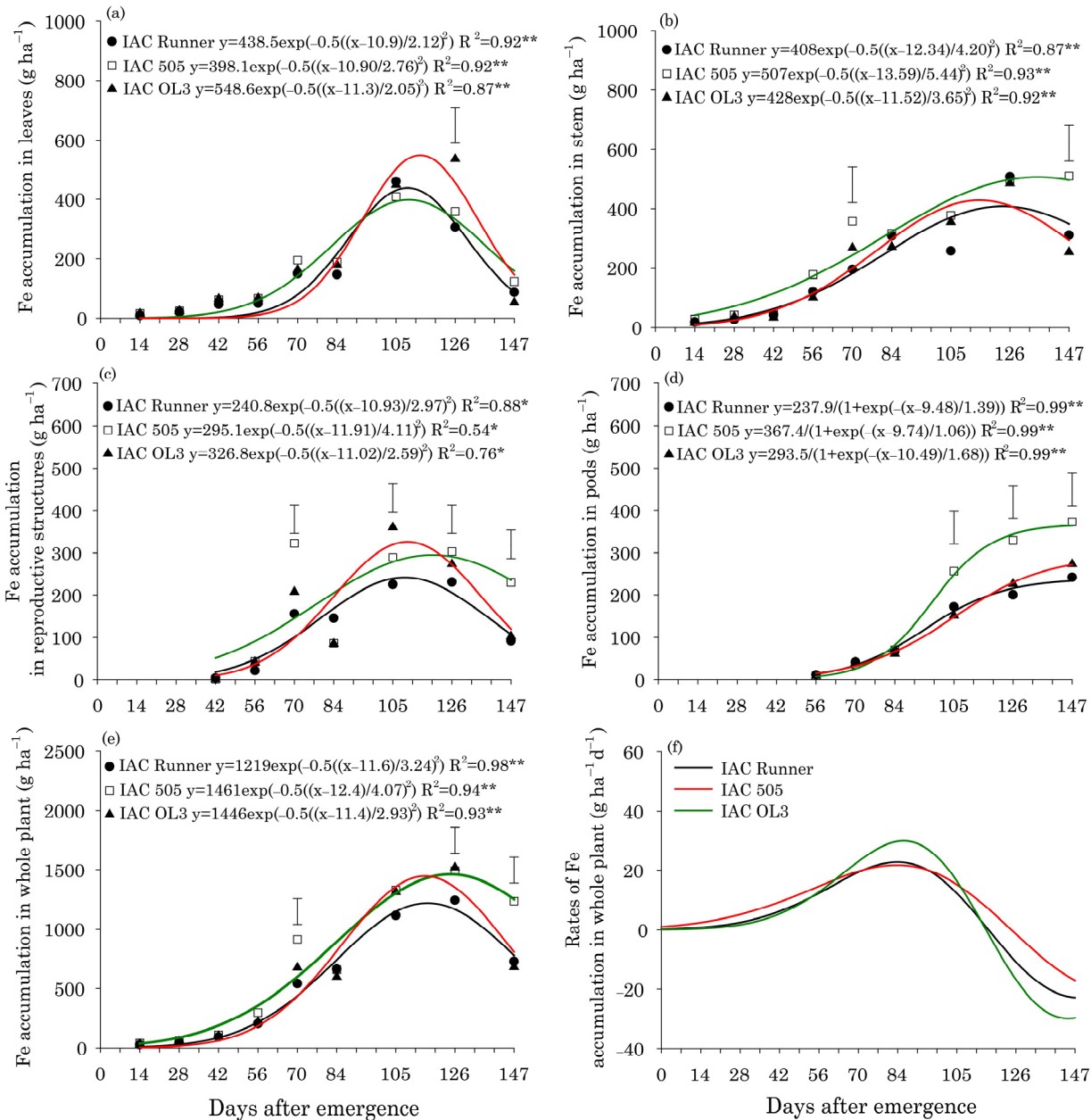

**Figure 3.** Iron (Fe) accumulation in leaves (**a**), stems (**b**), reproductive structures (**c**), pods (**d**), and the whole plant (**e**), and Fe accumulation rates in the whole plant (**f**) of three peanut cultivars throughout the season. ** and * indicate significance at $p \leq 0.01$ and $p \leq 0.05$ by the F test. The vertical bars indicate the least significant difference values by the LSD test at $p \leq 0.05$.

Manganese accumulation in leaves reached a maximum around 96 DAE in all peanut cultivars and was lowest in IAC Runner (Figure 4a). Mn accumulation in stems, reproductive structures, and the whole plant was highest at approximately 107 DAE in IAC Runner and IAC OL3, but at approximately 115 DAE in IAC 505 (Figure 4b,c,e). Between 105 and 147 DAE, IAC 505 showed superior Mn accumulation in stems, reproductive structures and in the whole plant compared with the other cultivars. Mn accumulated rapidly in peanut pods between 84 and 105 DAE, particularly in IAC 505 and IAC OL3 (Figure 4d). From 105 DAE until the end of the season (147 DAE), Mn accumulation in pods was significantly greater in IAC 505 than in the other cultivars. The rate of Mn accumulation in the whole plant was highest at around 77 DAE, and was higher in IAC 505 than in the other cultivars (Figure 4f).

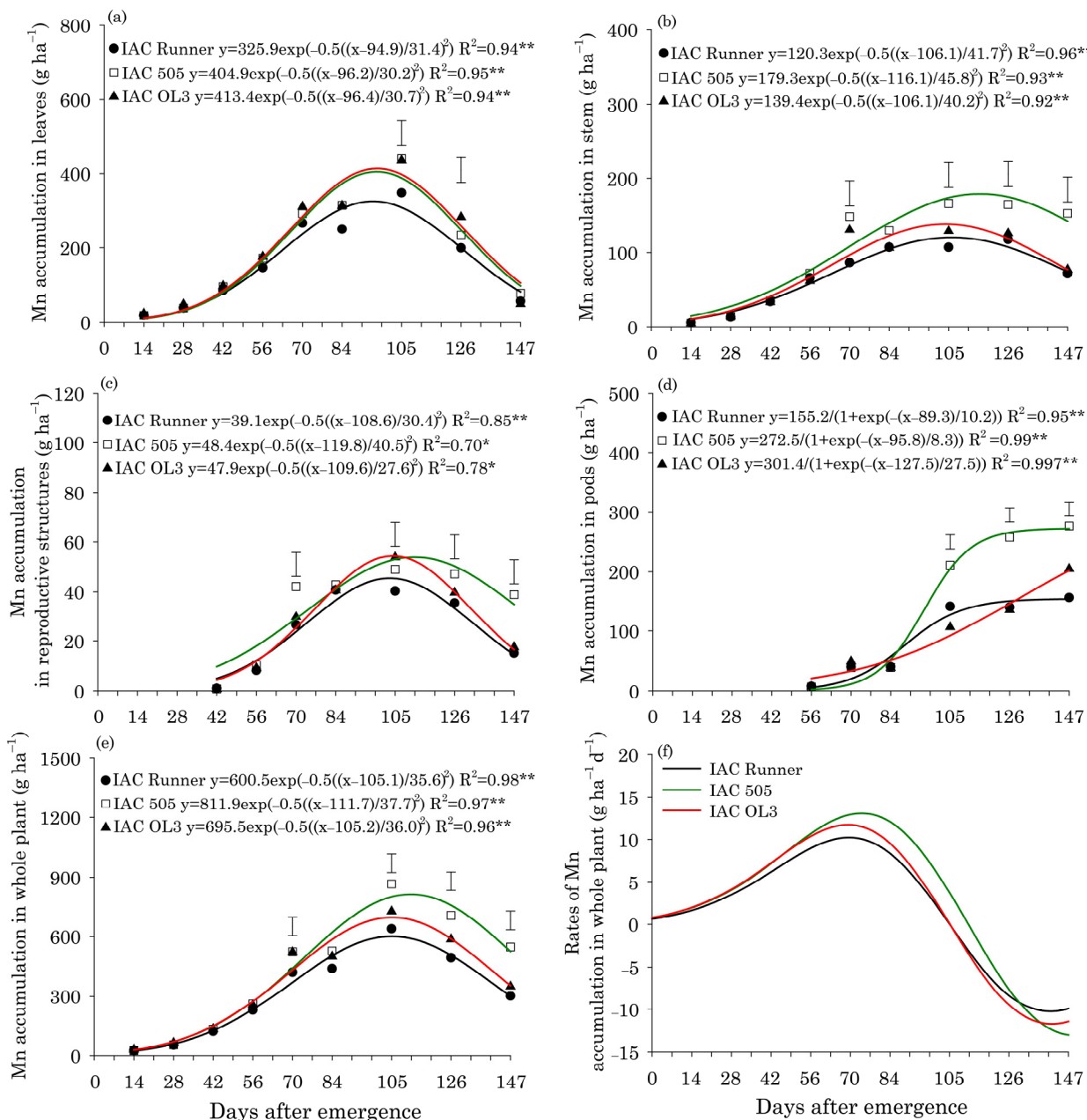

**Figure 4.** Manganese (Mn) accumulation in leaves (**a**), stems (**b**), reproductive structures (**c**), pods (**d**), and the whole plant (**e**), and Mn accumulation rates in the whole plant (**f**) of three peanut cultivars throughout the season. ** and * indicate significance at $p \leq 0.01$ and $p \leq 0.05$ by the F test. The vertical bars indicate the least significant difference values by the LSD test at $p \leq 0.05$.

Maximum Zn accumulation in peanut leaves occurred at approximately 85 DAE in all cultivars (Figure 5a). Zn accumulation in stems and reproductive structures peaked at 98 DAE in IAC Runner and IAC OL3, but ten days later in IAC 505, i.e., 108 DAE (Figure 5b,c). Zn accumulation in stems and reproductive structures was highest in IAC 505 at 105 and 147 DAE, respectively. In peanut pods, Zn accumulated rapidly in all cultivars after 84 DAE. However, Zn accumulation in pods was more pronounced in IAC 505 than in IAC Runner and IAC OL3 between 105 and 147 DAE (Figure 5d). The maximum Zn accumulation in the whole plant occurred at approximately 110 DAE in IAC Runner and IAC OL3, but at approximately 122 DAE in IAC 505 (Figure 5e). Zn accumulation in the whole plant was significantly higher in IAC 505 than in the other cultivars at 105, 126 and

147 DAE. The rate of accumulation of Zn in the whole plant was highest around 70 DAE in IAC Runner and IAC OL3 and around 80 DAE in IAC 505 (Figure 5f).

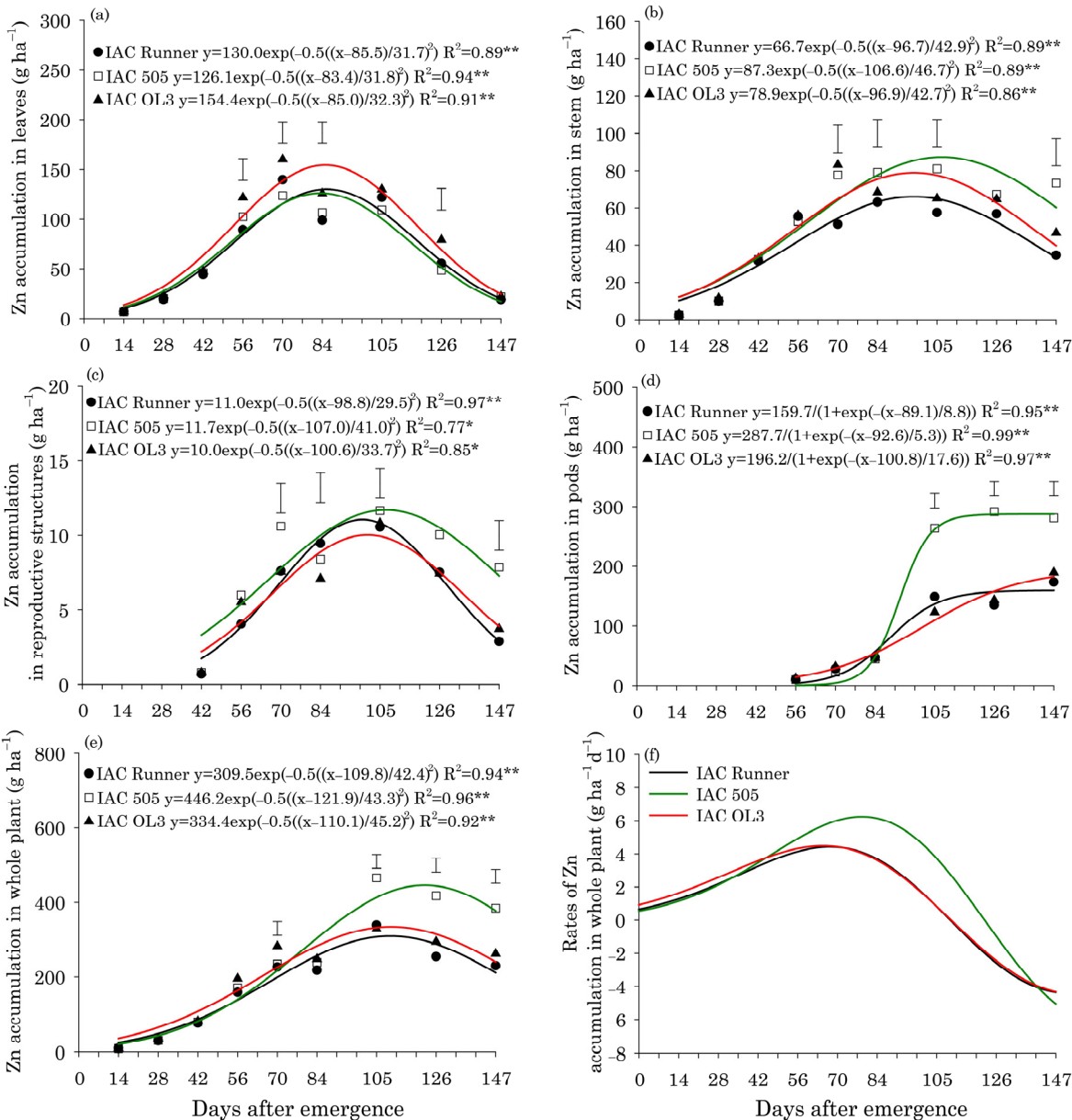

**Figure 5.** Zinc (Zn) accumulation in leaves (**a**), stems (**b**), reproductive structures (**c**), pods (**d**), and the whole plant (**e**), and Zn accumulation rates in the whole plant (**f**) of three peanut cultivars throughout the season. ** and * indicate significance at $p \leq 0.01$ and $p \leq 0.05$ by the F test. The vertical bars indicate the least significant difference values by the LSD test at $p \leq 0.05$.

Peanut pod and kernel yields were highest in IAC 505, followed by IAC OL3 and IAC Runner (Figure 6). The pod yield was 42% (2738 kg ha$^{-1}$) and 68% (3741 kg ha$^{-1}$) higher in IAC 505 than in IAC OL3 and IAC Runner, respectively.

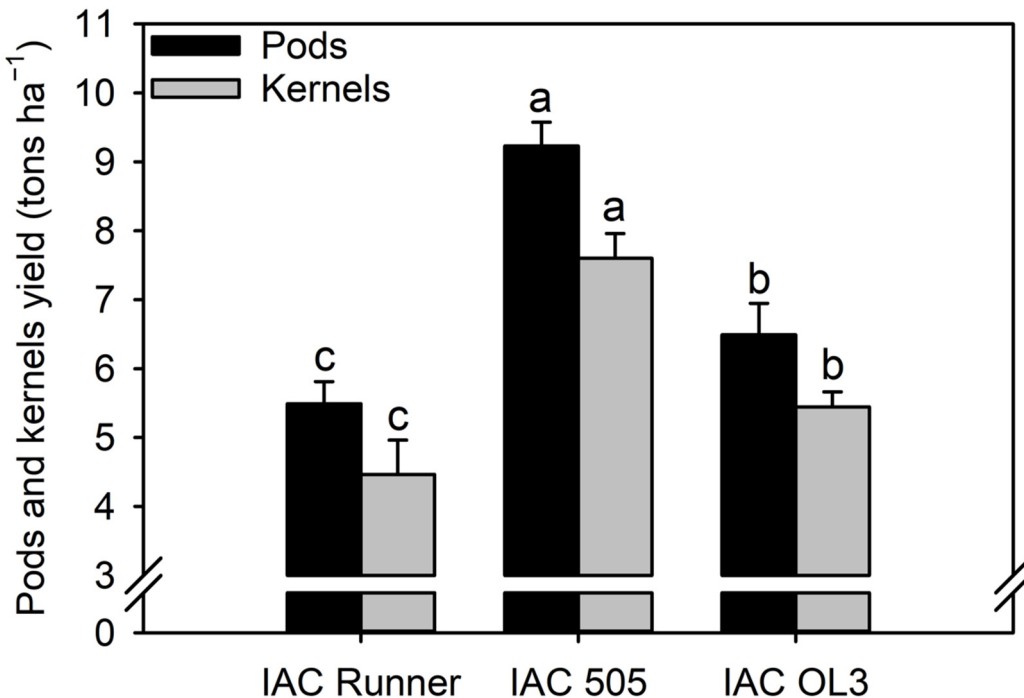

**Figure 6.** Pod and seed yields of the peanut cultivars. Different lowercase letters indicate significant differences between treatments (cultivars) by Fisher's protected LSD test at $p \le 0.05$.

Table 2 summarizes these findings. The uptake of B and Cu per ton of peanut pods and kernels produced was significantly greater in IAC Runner than in the other two cultivars (Table 2). IAC Runner accumulated 55.4 g of B and 30.4 g of Cu per ton of pods produced, and 68.1 g of B and 37.4 g of Cu per ton of kernels produced; compared to the other cultivars, IAC Runner absorbed 31.4% more B and 18.7% more Cu per ton of pods and kernels. The content of B in peanut pods was higher in IAC 505 than in IAC Runner. The Fe content was lowest in IAC 505 among all cultivars, and Mn content was higher in IAC OL3 than in IAC Runner (Table 2). With respect to the micronutrient content in kernels, only Mn content differed among the cultivars and was highest in IAC 505 (Table 2).

The uptake of micronutrients by peanut pods and kernels by area decreased in the following order: Fe > Zn > Mn > B > Cu (Table 2). In terms of removal by pods, IAC 505 stood out from the other cultivars, with average increases in the removal of Fe, Zn, Mn, B, and Cu of 45%, 55%, 53%, 66%, and 51%, respectively. The average removal of Fe per ton of pods produced was 6% lower in IAC 505 than in IAC Runner, and the average removal of Cu per ton of pods produced was 10% higher in IAC OL3 than in IAC Runner (Table 2). Compared with the other cultivars, IAC 505 also showed greater micronutrient removal by peanut kernels (56% Zn, 70% Mn, 72% B and 51% Cu). The removal of Fe by peanut kernels did not differ among the cultivars. Per ton of kernels produced, IAC 505 showed the greatest removal of B and Mn (Table 2). The relative removal of B, Fe, Mn, and Zn by pods (49.9%, 25.4%, 34.1%, and 63.0%, respectively) and kernels (41.4%, 2.1%, 18.4%, 57.7%, respectively) of IAC 505 was superior to those of the other cultivars (Table 2).

**Table 2.** Grams of micronutrients taken up per Mg of pods and kernels produced, micronutrient content in pods and kernels, micronutrient removal by pods and kernels per area, micronutrient removal per megagram (Mg) of pods and kernels produced, and relative micronutrient removal by pods and kernels of peanut cultivars.

| Cultivar | B | Cu | Fe | Mn | Zn |
|---|---|---|---|---|---|
| | Micronutrients taken up per Mg of pods produced (g Mg$^{-1}$) [1] | | | | |
| IAC Runner | 55.4 ± 1.8 a | 30.4 ± 0.8 a | 224.1 ± 44.9 a | 105.6 ± 10.2 a | 56.5 ± 4.5 a |
| IAC 505 | 37.8 ± 2.1 c | 25.9 ± 0.5 b | 158.9 ± 38.2 a | 87.8 ± 9.3 a | 48.4 ± 5.2 a |
| IAC OL3 | 47.5 ± 1.7 b | 25.9 ± 0.4 b | 222.6 ± 42.4 a | 104.7 ± 9.8 a | 51.6 ± 3.8 a |
| | Micronutrients taken up per Mg of kernels produced (g Mg$^{-1}$) [1] | | | | |
| IAC Runner | 68.1 ± 4.8 a | 37.4 ± 2.3 a | 275.8 ± 40.1 a | 129.9 ± 10.3 a | 69.5 ± 5.3 a |
| IAC 505 | 45.9 ± 5.7 b | 31.4 ± 3.8 ab | 193.0 ± 49.2 a | 106.5 ± 13.1 a | 58.8 ± 6.2 a |
| IAC OL3 | 56.6 ± 5.4 b | 30.9 ± 2.1 b | 265.3 ± 35. 3 a | 124.8 ± 9.8 a | 61.6 ± 4.4 a |
| | Micronutrient content in pods (mg kg$^{-1}$) | | | | |
| IAC Runner | 16.8 ± 0.6 b | 17.6 ± 0.5 a | 43.7 ± 0.9 a | 28.7 ± 0.6 b | 31.5 ± 1.2 a |
| IAC 505 | 18.9 ± 0.6 a | 17.3 ± 0.4 a | 40.1 ± 0.5 b | 30.0 ± 0.8 ab | 30.5 ± 0.8 a |
| IAC OL3 | 18.1 ± 0.8 ab | 17.8 ± 0.3 a | 41.9 ± 1.0 a | 31.5 ± 0.7 a | 29.2 ± 1.0 a |
| | Micronutrient content in kernels (mg kg$^{-1}$) | | | | |
| IAC Runner | 16.4 ± 1.5 a | 15.6 ± 0.9 a | 4.6 ± 1.1 a | 17.6 ± 0.5 b | 35.4 ± 2.3 a |
| IAC 505 | 19.1 ± 1.3 a | 15.2 ± 1.1 a | 4.0 ± 0.8 a | 19.7 ± 0.7 a | 33.9 ± 1.9 a |
| IAC OL3 | 17.4 ± 0.9 a | 15.4 ± 1.2 a | 3.4 ± 1.2 a | 18.0 ± 0.6 b | 31.6 ± 2.0 a |
| | Micronutrient removal by pods per area (g ha$^{-1}$) | | | | |
| IAC Runner | 92.7 ± 4.5 c | 96.7 ± 10.4 b | 240.9 ± 17.8 b | 156.9 ± 15.4 c | 173.5 ± 11.5 b |
| IAC 505 | 174.2 ± 5.1 a | 160.1 ± 9.2 a | 373.2 ± 15.2 a | 276.5 ± 19.1 a | 281.5 ± 17.2 a |
| IAC OL3 | 117.3 ± 5.2 b | 115.7 ±10.1 b | 272.7 ± 12.1 ab | 204.9 ± 10.1 b | 189.6 ± 12.3 b |
| | Micronutrient removal by kernels per area (g ha$^{-1}$) | | | | |
| IAC Runner | 73.4 ± 6.9 c | 69.3 ± 5.4 c | 22.4 ± 6.4 a | 77.6 ± 7.8 c | 158.2 ± 7.4 b |
| IAC 505 | 144.5 ± 8.3 a | 115.2 ± 9.8 a | 30.1 ± 6.9 a | 149.4 ± 9.3 a | 257.8 ± 8.5 a |
| IAC OL3 | 94.8 ± 7.3 b | 83.5 ± 6.1 b | 18.5 ± 7.3 a | 97.8 ± 6.4 b | 171.4 ± 9.9 b |
| | Micronutrient removal per Mg of pods produced (g Mg$^{-1}$) | | | | |
| IAC Runner | 16.8 ± 2.4 a | 17.6 ± 0.5 a | 43.7 ± 1.4 a | 28.7 ± 0.8 b | 31.5 ± 0.9 a |
| IAC 505 | 18.9 ± 2.1 a | 17.3 ± 0.7 a | 40.1 ± 0.5 b | 29.9 ± 0.4 ab | 30.5 ± 1.0 a |
| IAC OL3 | 18.1 ± 1.8 a | 17.8 ± 0.3 a | 41.9 ± 0.9 a | 31.5 ± 0.5 a | 29.2 ± 1.4 a |
| | Micronutrient removal per Mg of kernels produced (g Mg$^{-1}$) | | | | |
| IAC Runner | 16.4 ± 0.5 c | 15.6 ± 0.5 a | 4.6 ± 0.6 a | 17.6 ± 0.5 b | 35.4 ± 2.1 a |
| IAC 505 | 19.1 ± 0.4 a | 15.2 ± 0.6 a | 4.0 ± 0.4 a | 19.7 ± 0.7 a | 33.9 ± 1.5 a |
| IAC OL3 | 17.4 ± 0.4 b | 15.4 ± 0.4 a | 3.4 ± 0.6 a | 18.0 ± 0.6 b | 31.6 ± 1.7 a |
| | Relative micronutrient removal by pods (%) [2] | | | | |
| IAC Runner | 30.5 | 57.9 | 19.6 | 27.1 | 56.0 |
| IAC 505 | 49.9 | 67.0 | 25.4 | 34.1 | 63.0 |
| IAC OL3 | 38.1 | 68.9 | 18.9 | 30.2 | 56.6 |
| | Relative micronutrient removal by kernels (%) [2] | | | | |
| IAC Runner | 24.2 | 41.5 | 1.8 | 13.4 | 51.0 |
| IAC 505 | 41.4 | 48.2 | 2.1 | 18.4 | 57.7 |
| IAC OL3 | 30.7 | 49.7 | 1.3 | 14.4 | 51.1 |

Values ± SE (standard error) followed by the same letter in the same column within each parameter are not significantly different at $p \leq 0.05$ according to the LSD test. [1] Data are based on pod and kernel yields (Figure 6) and on the values of maximum micronutrient accumulation in the whole plant as shown in Figures 1e, 2e, 3e, 4e and 5e. [2] Proportional micronutrient removal by pods (pods + kernels) or kernels in relation to the maximum micronutrient uptake by three peanut cultivars as shown in Figures 1e, 2e, 3e, 4e and 5e.

## 4. Discussion

*Concentrations and Uptake of Micronutrients*

In this study of modern cultivars with greater productive potential, the leaf contents of Fe, Mn, B, Zn, and Cu were within the ranges considered appropriate for peanut in all cultivars [39] (Fe: 50 to 300 mg kg$^{-1}$; Mn: 20 to 350 mg kg$^{-1}$; B: 25 to 60 mg kg$^{-1}$; Zn: 20 to 60 mg kg$^{-1}$; Cu: 17 to 20 mg kg$^{-1}$). These results indicate that the micronutrient availability in the soil, which was medium to high, was not limiting for these modern

cultivars [40]. Among the evaluated micronutrients, the content of Fe was highest in peanut leaves, followed by Mn, B, Zn, and Cu. A previous study of older peanut cultivars (Oirã, Poitara, Tupã and Penápoles) [41] reported that total micronutrient accumulation in peanut plants decreased in the order Fe > Cu > Zn > B > Mn; by contrast, in the present study of new cultivars, the pattern was Fe > Mn > Zn > B > Cu. This order was observed for micronutrient transport to pods and seeds production as well as by area. Compared with the old cultivars, the new cultivars accumulated more Mn and less Cu. Micronutrient removal by area was highest in IAC 505, followed by IAC OL3 and IAC Runner 886, the same order in which these cultivars produced dry biomass. Moreover, consistent with previous findings that Fe is the least translocated element to pods and kernels [41], Fe in pods and kernels corresponded to only 21% and 1.73%, respectively, of the total Fe absorbed. The transport of nutrients to pods and seeds decreased in the order Zn > Cu > B > Mn > Fe. Therefore, during sugarcane renovation, modern peanuts cultivars must be fertilized with Zn, Cu, and B because these cultivars export approximately half of the absorbed quantities of these micronutrients to pods, which can deplete the micronutrient content of the soil.

Manganese is involved in several physiological processes in higher plants, particularly as an enzymatic activator of enzymes such as Mn-containing superoxide dismutase (Mn-SOD) and oxalate oxidase (OXO; EC 1.2.3.4) [42]. Mn is also involved in water photolysis by photosystem II [43], which provides the electrons needed for photosynthesis [44]. The greater accumulation of Mn in new peanut cultivars may reflect greater tolerance to the presence of Mn. Tolerance to Mn toxicity is an important physiological attribute in crops grown in acid soils. Approximately half of the world's arable soils have low pH and excess $Al^{3+}$ and Mn content [45]. Mn content in plant tissues varies between species and genotypes [46]. Plants with greater Mn absorption capacity and use efficiency will achieve greater development and yields under suboptimal Mn availability, mainly by providing sufficient Mn for photosystem II and increasing photosynthetic efficiency [47].

As expected, micronutrient accumulation followed the same pattern as dry matter accumulation [34], with maximum values occurring at 105 DAE in IAC Runner and IAC OL3 and approximately 125 DAE in IAC 505. The temporal profile of micronutrient absorption revealed that the requirement for Fe, B, Cu, Mn, and Zn ($g\,ha^{-1}\,d^{-1}$) was highest from emergence until 70–84 DAE, and declined thereafter. In IAC 505, micronutrients were absorbed at faster rates and at a later stage of the cycle than in the other cultivars. The higher micronutrient absorption and removal of IAC 505 is a result of its high biomass production [34] and high pod and kernel yields (Figure 6). The micronutrient requirements in the initial period ($g\,ha^{-1}\,d^{-1}$) were approximately 20–30 Fe, 4.5 B, 1.9–2.2 Cu, 9–13 Mn, and 4–6 Zn. These results provide a greater understanding of the period of greatest need for micronutrients during the peanut crop cycle, which will allow proper fertilization (via soil or foliar applications) to be carried out when needed.

## 5. Conclusions

The modern peanut cultivars assessed in this study showed a high uptake and accumulation of Fe in all plant tissues, although the proportion of Fe removed by pods and kernels was the lowest among the analyzed micronutrients. Mn was the second-most-accumulated micronutrient by modern peanut cultivars. The rates of micronutrient uptake ($d^{-1}\,ha^{-1}$) by the peanut cultivars peaked at around 80 DAE, and IAC 505 had higher micronutrient uptake rates than the other cultivars at later stages of the cycle. Among the cultivars, IAC 505 had the highest requirement for micronutrients, followed by IAC OL3 and IAC Runner 886. Our results demonstrate the importance of updating fertilization recommendation programs for peanut crops to include micronutrients. Our study provides essential information on the periods in which micronutrients should be supplied to the crop if needed. Finally, our results clarify the dynamics of micronutrient uptake by modern peanut cultivars and provide a much-needed reference for future research, seeking to increase the yield and nutritional quality of peanut kernels by applying adequate amounts of micronutrients.

**Supplementary Materials:** The following supporting information can be downloaded at https://www.mdpi.com/article/10.3390/crops3020010/s1. Table S1: Rainfall, maximum and minimum temperatures at Botucatu, São Paulo, Brazil, during the study period and averages.

**Author Contributions:** Conceptualization, C.A.C.C., A.M.F. and H.C.; methodology, L.G.M., J.R.P. and J.W.B.; software, J.W.B. and C.P.; validation A.M., C.P. and J.R.P.; formal analysis, J.W.B., L.G.M., J.L.N.G. and G.L.d.B.G.; investigation, J.W.B.; resources, C.A.C.C.; writing—original draft preparation, H.C.; writing—review and editing, J.W.B. and J.R.P.; supervision, C.A.C.C. and H.C. All authors have read and agreed to the published version of the manuscript.

**Funding:** The first author would like to thank the National Council for Scientific and Technological Development (CNPq) for an award for excellence in research (grant number: 303119/2016-0).

**Data Availability Statement:** The datasets analyzed during the current study are available from the corresponding author upon reasonable request.

**Acknowledgments:** The National Council for Scientific and Technological Development (CNPq) is acknowledged for the "Excellence in Research" award given to the first author. The authors would like to thank COPERCANA (Cooperative of Sugarcane Farmers in the West of São Paulo State) for support during the experiment.

**Conflicts of Interest:** The authors declare no conflict of interest.

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
