# Peer review of "Dynamics of Micronutrient Uptake and Removal by Three Modern Runner Peanut Cultivars"

_2673-7655, doi:10.3390/crops3020010_

Round 1

Reviewer 1 Report

Dear authors,

Minor errors in the study  were shown and corrected.

The discussion part of the study can be strengthened a little more.

With best wishes...

Author Response

The authors would like to thank the reviewer for such a great and positive review. The comments and suggestions provided by you have made this manuscript much better.  All corrections were made in the final file.

Reviewer 2 Report

This paper reports the micronutrient data obtained from a field experiment from which the macronutrient data have already been published separately. The experiment was not designed to explore needs for micronutrient fertilisation, and the discussion is at times misleading on this point.

The justification for this trial was the need to update information based on older cultivars, given the greater growth and yield potential of new prostrate cultivars. However, only “new” cultivars were included in the trial.

I resented having to retrieve and read the companion article “Dynamics of Macronutrient Uptake and Removal by Modern Peanut Cultivars” in order to obtain basic information about plot design, sampling and analysis. All the information for evaluating this study should be included in this paper, regardless of whether it was published in the earlier paper. Even there, the information was insufficient to explain how the sampling regime affected the space and nutrients available to the remaining plants.

The usefulness of the many charts on nutrient accumulation is questionable, since it is not related to the dry matter accumulation (which was only published in the companion paper). It is very evident that the nutrient accumulation was closely linked to dry matter accumulation. The greater growth and yield of IAC 505 in this particular trial fully accounts for its greater nutrient accumulation, but this point is not acknowledged. Instead we are treated to a cookie-cutter repetition of the description of nutrient uptake for each nutrient. This information is hardly generalizable, when the description of cultivars (given only in the companion article) reports all as having similar yield potential. Statements such as “Between 105 and 147 DAE, IAC 505 outperformed the other cultivars in B accumulation in the whole plant” (L 180) are repeated for every nutrient, without making the obvious point that the accumulation of all the nutrients followed dry matter accumulation, and, in this particular trial, IAC 505 grew more than the others. In the discussion, at L 318, the authors note as if by coincidence, “the same order in which these cultivars produced dry biomass.” The importance of this for the generalisability of their data is not noted.

Regarding Table 1, no discussion is offered why the full-bloom apical cluster samples were taken and analysed, in addition to the entire above-ground parts. Clearly this is for comparison with critical nutrient concentrations, according to a standardised crop sampling regime. At the start of the discussion, these concentrations are compared with “those considered appropriate” so that it is established that none of the cultivars experienced a deficiency of these nutrients. Having then said this “indicates that the micronutrient availability in the soil was not limiting for these modern cultivars, since the levels in the soil were medium to high” (L308-309) I was surprised to read “fertilization of modern peanuts cultivars with Zn, Cu, and B must be carried out because these cultivars remove with the pods approximately half of the absorbed quantities, which can deplete the micronutrient soil contents.” (L 322-324) Micronutrient fertilisation is not normally recommended unless a soil deficiency is expected.

To illustrate: even for Zn, the micronutrient with the lowest concentration measured in soil analysis, 2.6 mg dm-3 is equal to 5.2 kg ha-1 in the top 20 cm, counting only the readily extractable fraction. In contrast, the results in Table 2 report Zn uptake by the whole plant tops to be about a third of one kg per ha and that removed in pods about 0.2 kg ha-1. Given the reserves nutrients more tightly bound to the soil matrix, it is unclear whether micronutrient supplementation would ever be necessary.

There are many minor errors in the text, many of which I have highlighted in the annotated manuscript.

I find the paper underwhelming. It is reporting data from a competently run field trial, but the usefulness and generalisability of that data is questionable, and the justifications given in the introduction and conclusion seem contrived. The Conclusion announces, “Our results demonstrate the importance of updating fertilization recommendation programs for peanut crops to include micronutrients.” (L 359) But previously published information on nutrient uptake is not cited. I imagine there would be little change to previous estimates of nutrient removal per tonne of pods, with the difference between these cultivars and earlier ones being accounted for in their yield. To the extent that the data are useful, the nutrient concentration in the apical cluster should have been reported briefly in the earlier paper, if only to demonstrate the validity of that paper’s macronutrient results, by confirming that the plants did not suffer any micronutrient undernourishment. Beyond that, including the micronutrients in a tabulation of nutrient removal per tonne of pods would have given all the necessary information. Instead the authors have sought to get two papers from the one trial on thin justification. It is not a strategy that I think should be rewarded.

Author Response

We appreciate your comments, and we've tried our best to revise the article.

Reviewer 3 Report

I have read the manuscript entitled “Dynamics of micronutrient uptake and removal by modern peanut cultivars”. This is an excellent manuscript that reports in this study for the demand for micronutrients (B, Cu, Fe, Mn, and Zn) by three runner peanut cultivars (IAC Runner 886, IAC 505 and IACOL3) during the biological cycle, as well as the export of these micronutrients to pods and kernels. Generally, I believe the manuscript can interest for the journal. Although, I am satisfied with the overall presentation and writing of the manuscript. I would suggest minor revision and give the chance to the authors to overlook the following remaining errors in the manuscript.

1.      Line 30: In nine plant samplings, I observed till 70 day after emergence your sampling time is going smoothly as 14 days intervals why suddenly changes after that any logical reason or it does not impact on the demand of nutrients.

2.      Iine 34: 80 DAE is not in your plant sampling, check it.

3.      When was your lowest requirement or uptake for micronutrients by peanuts cultivars?

4.      How about Mn and Zinc response in peanut cultivars?

Author Response

(The authors gave the same response as above.)

Reviewer 4 Report

Dear Authors,

you have taken up the important topic of plant nutrition with micronutrients.
Your research is important because we often rely on old recommendations
not adapted to new varieties.

Methodology

Please specify the method for determination of boron. This is not an ASA method.

Line 27 : as well as the tranport export of these micronutrients to pods and kernels

Line 30. Double bracket

Word export replace with a more suitable one transport

Many places; reproductive structures the botanical name of this part peanut is peg

Line Figure 6. Term kernel is colloquial, botanically it is a seed

Line 88 ……………respiratory process [29]. If you describe all micronutrients, mention chlorine in the context of photosynthesis. After all, we still have nickel on the list, but we rarely mention it.

Lines 89-99. Ad some recommendations for molybdenum.

Line 281 The removal of micronutrients by peanut pods and kernels by area decreased in the 281 following order.  More suitable is  uptake than removal

Line 273. IAC Runner absorbed 55.4 g … it will be more corret IAC accumulated 55.4 g…

Line 281. Replace removal by uptake. The same sytuation in many places.

Line 321. Export of Nutrients transport to pods and seeds de-creased in the order Zn > Cu > B > Mn > Fe

Refernces

Generally latin names are written in itallics.

1.     Sharma, C.P. Plant Micronutrients; CRC Press: Boca Raton, 2006; ISBN 0429079427

Sharma, C.P. Plant Micronutrients; CRC Press: Boca Raton, 2006, ISBN 0429079427.

5. CONAB Acompanhamento Da Safra Brasileira - Sétimo Levantamento Safra 2020/21; Brasília, DF, 2021; Vol. 8;.

28. Pandey, S.N. Role of Micronutrients in Biochemical Responses of Crops Under Abiotic Stresses; Springer, 2020;……………….????

Ref. 35-40. Many mistakes, localization, bold, dots … Repare please.

Author Response

(The authors gave the same response as above.)

Round 2

Reviewer 2 Report

The revised manuscript has not addressed any of the issues I raised with the original version. The study was not designed to test the need for, or value of, micronutrient fertilisation. The methodology still requires the reader to consult another paper for key information. The results still present nutrient removal as if it is not a function of biomass. Comments such as "Therefore, during sugarcane renovation, peanuts cultivars must be fertilized with Zn, Cu, and B must be carried out because these cultivars export approximately half of the absorbed quantities of these micronutrients to pods, which can deplete the micronutrient content of the soil" are inconsistent with the soil test results provided. The paper does not compare the new cultivars with older cultivars. It provides no novel information that is generalisable and useful to crop managers.